# Engineered symbiotic bacteria interfering *Nosema* redox system inhibit microsporidia parasitism in honeybees

Haoyu Lang [1], Hao Wang[1], Haoqing Wang[1], Zhaopeng Zhong[1], Xianbing Xie[2], Wenhao Zhang[3], Jun Guo[4], Liang Meng[5], Xiaosong Hu[1], Xue Zhang [6] & Hao Zheng [1] ✉

*Nosema ceranae* is an intracellular parasite invading the midgut of honeybees, which causes serious nosemosis implicated in honeybee colony losses worldwide. The core gut microbiota is involved in protecting against parasitism, and the genetically engineering of the native gut symbionts provides a novel and efficient way to fight pathogens. Here, using laboratory-generated bees mono-associated with gut members, we find that *Snodgrassella alvi* inhibit microsporidia proliferation, potentially via the stimulation of host oxidant-mediated immune response. Accordingly, *N. ceranae* employs the thioredoxin and glutathione systems to defend against oxidative stress and maintain a balanced redox equilibrium, which is essential for the infection process. We knock down the gene expression using nanoparticle-mediated RNA interference, which targets the γ-glutamyl-cysteine synthetase and thioredoxin reductase genes of microsporidia. It significantly reduces the spore load, confirming the importance of the antioxidant mechanism for the intracellular invasion of the *N. ceranae* parasite. Finally, we genetically modify the symbiotic *S. alvi* to deliver dsRNA corresponding to the genes involved in the redox system of the microsporidia. The engineered *S. alvi* induces RNA interference and represses parasite gene expression, thereby inhibits the parasitism significantly. Specifically, *N. ceranae* is most suppressed by the recombinant strain corresponding to the glutathione synthetase or by a mixture of bacteria expressing variable dsRNA. Our findings extend our previous understanding of the protection of gut symbionts against *N. ceranae* and provide a symbiont-mediated RNAi system for inhibiting microsporidia infection in honeybees.

Honeybees (*Apis mellifera*) are pollinators with global economic value responsible for pollinating ecologically and agriculturally valuable crops. For the past decade, a phenomenon known as colony collapse disorder has posed a global threat to honeybee health. Recent studies suggest several factors involved in colony decline, such as parasite and pathogen invasion, pesticide use, and environmental stressors. Honeybees are susceptible to various pathogens and pests, including bacteria[1], fungi[2], viruses[3], *Varroa* destructor[4], and microsporidian parasites[5].

[1]College of Food Science and Nutritional Engineering, China Agricultural University, 100083 Beijing, China. [2]Department of Laboratory Animal Science, Nanchang University, 330006 Nanchang, China. [3]Faculty of Agriculture and Food, Kunming University of Science and Technology, 650031 Kunming, China. [4]Faculty of Life Science and Technology, Kunming University of Science and Technology, 650031 Kunming, China. [5]BGI-Qingdao, BGI-Shenzhen, 266555 Qingdao, China. [6]College of Plant Protection, China Agricultural University, 100083 Beijing, China. ✉e-mail: hao.zheng@cau.edu.cn

The microsporidia are obligate intracellular eukaryotic parasites of honeybees and infect the midgut epithelial cells. Honeybees are mainly infected by two species of microsporidia that cause nosemosis, one of the most severe bee diseases worldwide[6]. *Nosema apis* was initially described in European honeybees and was considered the exclusive parasite species causing nosemosis. Later, another species *Nosema ceranae* was discovered in the Asian honeybee, *Apis cerana*, which is presumed to be the original host. It may transfer to *A. mellifera* during the past decades[7]. It appears that *N. ceranae* displaces *N. apis* in *A. mellifera*, and the prevalence studies found that *N. apis* infections are becoming rarer than *N. ceranae*[8]. *N. ceranae* transmit via the fecal-oral route and the ingestion of spores from the contaminated hive materials[9]. It can suppress the immune defense mechanism of honeybees, ensuring the infection of epithelial cells[10,11]. The parasitic infection reduces the lifespan and colony populations of *A. mellifera* and affects host physiology and behaviors[12,13].

Honeybees rely on innate immunity to defend against infectious agents, which operate through cellular and humoral mechanisms[14]. The humoral immune system consists of antimicrobial peptide (AMP) production, primarily protecting against bacterial pathogens. For intracellular parasites, insects can clear invading parasites by eliciting oxidative stress[15–17]. The intestinal epithelial and macrophage cells produce reactive oxygen species (ROS)[18], including superoxide anion ($O_2^-$), hydrogen peroxide ($H_2O_2$), and hydroxyl radical (HO•). While there is no evidence that ROS is effective in clearing microsporidia, the infection of *N. ceranae* may disrupt the oxidative balance of the honeybee gut[19].

Host ROS production can be modulated by the gut microbiota to eliminate opportunistic pathogens[18]. Although it is unclear whether the microbiota inhibits the parasitism, *N. ceranae* infection perturbs the native gut composition, which may enhance the intensity of the parasitic microsporidia[20]. The honeybee gut microbiota typically contains five core bacterial members[21]. It has been shown that the bee gut bacteria influence bee health by modulating host immune responses. Specifically, *Snodgrassella alvi* and *Lactobacillus apis* protect honeybees from opportunistic bacterial pathogens by inducing host immune response and AMP production[22,23]. Furthermore, the native gut bacteria can be engineered to better improve honeybee health[24]. Leonard et al. recently genetically modified *S. alvi*, refining a system to induce RNAi within hosts. By expressing dsRNA to interfere gene expression of *Varroa* mite and DWV, the genetically engineered strains repress DWV and *Varroa* infection[25]. RNAi has been explored to discover novel targets and treatments for *N. ceranae* infections. Previous works have applied RNAi of variable targets, such as the ATP/ADP transporter[26], the polar tube protein (ptp)[27], and the spore wall protein (SWP)[28], to control nosemosis in honeybees. Thus, symbiont-mediated RNAi may provide a promising strategy for improving bee resistance against *Nosema* disease.

Here, we investigate the effect of honeybee gut members on the inhibition of *N. ceranae* invasion. Specifically, *S. alvi* upregulated the expression of host genes related to the ROS-associated immune response and significantly repressed the proliferation of *N. ceranae*. Then, we evaluated the role of the antioxidant system of *N. ceranae* in the adaptation and reproduction in the midgut epithelia. We found that *N. ceranae* mainly employed the thioredoxin and glutathione systems to relieve the intense oxidative stress from the host for parasitism. Finally, we constructed recombinant *S. alvi* to continuously produce dsRNA corresponding to the thioredoxin and glutathione system-related genes of *N. ceranae*, significantly inhibiting the *N. ceranae* proliferation in the midgut cells.

## Results

### Gut bacteria aid in the clearance of the pathogenic *N. ceranae*
We first determined whether the core gut members prevent the invasion of *N. ceranae* in vivo. We colonized microbiota-free (MF) bees with five core gut bacteria of honeybees, *Bifidobacterium choladohabitans* W8113, *Bombilactobacillus mellis* W8089, *L. apis* W8172, *Gilliamella apicola* B14384H2, and *S. alvi* M0351 in the lab (Supplementary Fig. 1). After allowing the colonization of symbiotic strains in the gut for 7 days, each bee individual was manually infected with *N. ceranae* cell suspensions of $10^4$ spores by oral feeding (Fig. 1a). On day 17, we quantified the absolute abundance of *N. ceranae* spores in the midguts. It showed that the spore load was significantly lowered in bees mono-colonized with *S. alvi*, while bees colonized by other gut members did not show a significant reduction of *N. ceranae* compared to the MF group (Fig. 1b and Supplementary Fig. 2).

Insects can clear parasites from invasion by eliciting oxidative stress, primarily by producing ROS in gut epithelia[29,30]. Thus, we assessed whether *S. alvi* stimulated the production of ROS in the gut, which may fight against invading intracellular *N. ceranae* parasite. In the honeybee, the production of ROS is mainly regulated by the Nox/Duox NADPH oxidases, as in other insects[31]. We found that the expression of genes encoding Duox and Nox were upregulated in the midgut of bees 24 h post-colonization by *S. alvi* M0351 (Fig. 1c, d). In contrast, the other gut bacteria did not upregulate the expression of the two genes (Supplementary Fig. 3). Correspondingly, both the intracellular ROS signal tested by the fluorogenic sensor and the production of hydrogen peroxide ($H_2O_2$) increased in the midgut following the colonization by *S. alvi* (Fig. 1e, f). These results indicate that the colonization of the core gut member, *S. alvi*, triggered the redox response involved in gut immunity, which may inhibit the *N. ceranae* infection in the honeybees.

### *N. ceranae* employs antioxidant systems to adapt and reproduce in the midgut epithelium
We have shown that ROS produced by the bees is implicated in the defense against *N. ceranae*, and typically, the parasites employ endogenous antioxidant systems to relieve intense oxidative stress[32,33]. To examine the pathways used by *N. ceranae* to resist honeybee gut oxidative stress during infection, we revisited an RNA-seq dataset that documents the changes in gene expression of *N. ceranae* when colonizing the bee gut. It probed the gene expression of 15 pooled bee individuals from three cup cages during the 6-day infection, but it should be noted that this dataset was achieved without replication[34]. De novo synthesis of reduced glutathione synthesized by γ-glutamyl-cysteine synthetase (γGCS) and glutathione synthetase (GS) is crucial in the antioxidant defense of *N. ceranae* (Fig. 2a). By following the time-series gene expression profiles, we found that both the expression of *γGCS* and *GS* of *N. ceranae* increased along with the infection (Fig. 2b, c). Moreover, glutathione can be further catalyzed by the glutathione peroxidases (GPx) to reduce $H_2O_2$[35]. We identified that *N. ceranae* possessed two genes encoding *GPx* in the genome of *N. ceranae*, *GPx*−1(AAJ76_3500027152) and *GPx*−2 (AAJ76_3500027978). Interestingly, the expression of *GPx*-1 increased during the first 3 days of infection, but *GPx*-2 was downregulated during invasion (Fig. 2d, e). In addition, *N. ceranae* also possesses a complete thioredoxin system, consisting of the key enzymes of thioredoxin reductase (TrxR, AAJ76_5800012528) and thioredoxin peroxidase (TPx, AAJ76_280004776), in defense against oxidative stress (Fig. 2f, g). We found that the expression of *TrxR* and *TPx* genes of *N. ceranae* were elevated from day 2 post-infection (Fig. 2h, i).

To further validate the importance of the thioredoxin and glutathione redox systems for the *N. ceranae* invasion, we knocked down *γGCS* from the glutathione system and *TrxR* from the thioredoxin system, respectively. Here, we used the nanoparticle-mediated dsRNA delivery system to improve RNAi efficiency (Fig. 2j)[36]. By feeding the nanoparticle-mediated dsRNA to MF bees, the mRNA transcript levels of *N. ceranae* γGCS and *TrxR* genes were reduced by ~80% on day 10 and day 15 after inoculation (Fig. 2k, i). Microscopic observation confirmed that

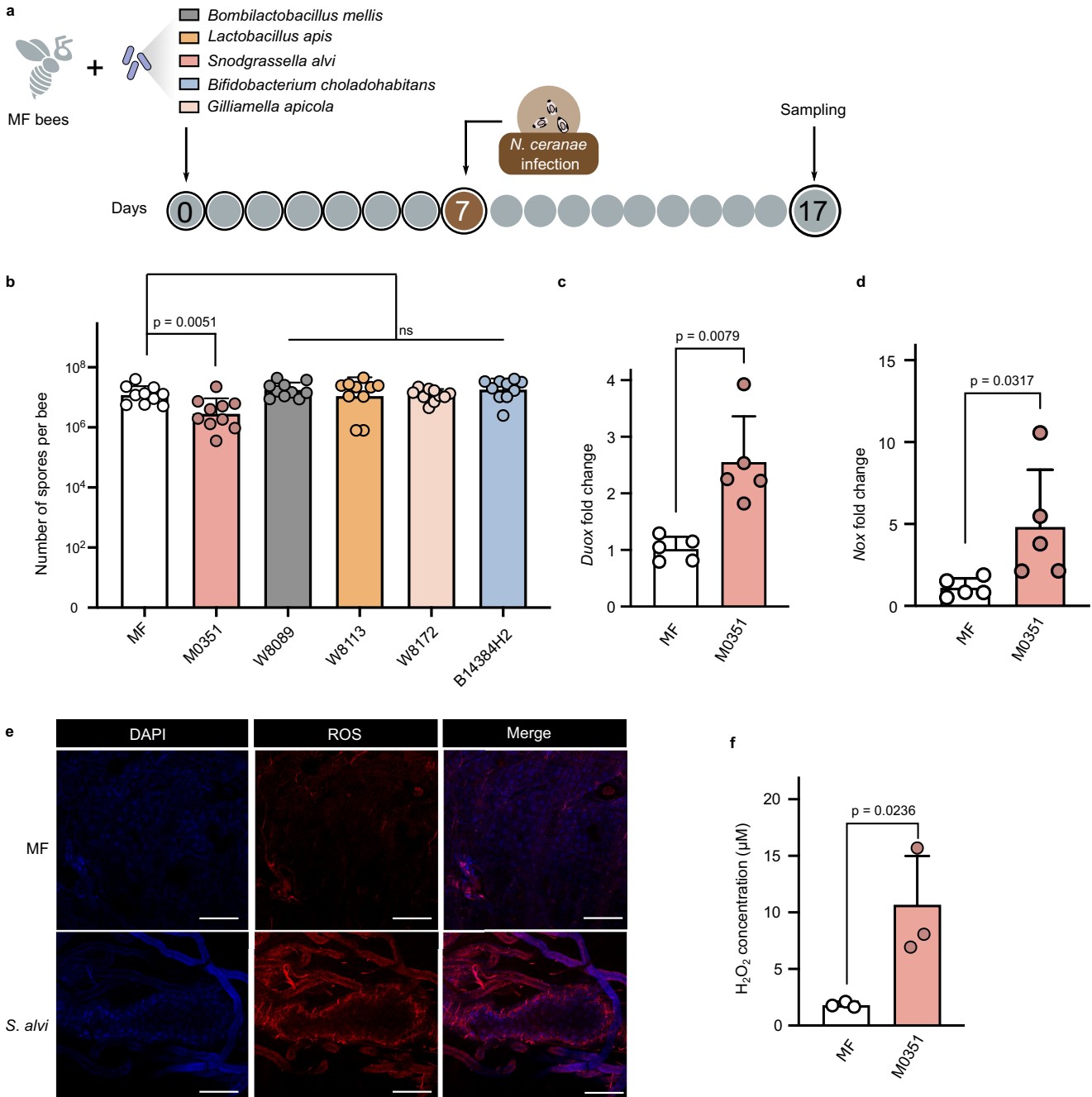

**Fig. 1 | *Snodgrassella* strains protect against *N. ceranae* via the ROS-associated immune response in the honeybee gut. a** Schematic illustration of experimental design. Microbiota-free (MF) bees were colonized with *B. choladohabitans* W8113, *B. mellis* W8089, *L. apis* W8172, *G. apicola* B14384H2, and *S. alvi* M0351 for 7 days and then orally infected with *N. ceranae*. See also Supplementary Fig. 1. **b** Absolute abundance of *N. ceranae* spores in the midgut 10 days post-infection with *N. ceranae*. *n* = 10 bees from two cup cages. The pairwise difference between the MF group and each mono-colonization group was tested by the two-sided Mann–Whitney *U* test. **c, d** The expression level of the *Duox* and *Nox* genes in the midgut following *S. alvi* M0351 colonization (*n* = 5 bees for both groups). Two-sided Mann–Whitney *U* test. **e** Fluorescence staining for Reactive oxygen species (ROS) signal (Red) within the midgut cells of MF and mono-colonized bees with *S. alvi*. Honeybee gut stained using 4′,6-diamidino-2-phenylindole (DAPI; blue). **f** H$_2$O$_2$ concentration in the midgut of MF and mono-colonized bees with *S. alvi* (*n* = 3 bees for both groups). Scale bars = 250 µm. Multiple two-tailed *t*-tests. Error bars represent mean SD. Source data are provided as a Source Data file.

the proliferation of *N. ceranae* was significantly depressed by both dsγGCS and dsTrxR silencing in the midgut (Fig. 2m–o). Altogether, these results indicate that *N. ceranae* probably maintains the redox state by employing the thioredoxin and glutathione systems to relieve the oxidative stress from the host and to adapt and reproduce in the midgut epithelium[37]. This also implies that host ROS-associated immunity is responsible for the defense against intracellular parasitism in honeybees.

### Inhibition of *N. ceranae* infection by engineered *S. alvi*

Since the antioxidant defense is crucial for *N. ceranae* parasitism, we next engineered *S. alvi* strain M0351 to produce dsRNA targeting microsporidian genes. First, we transformed strain M0351 with a stable plasmid pBTK501 expressing *GFP* from the Bee Microbiome Toolkit platform[24] and tested whether it could re-colonize bee gut robustly. No changes in morphology or growth rate for the engineered M0351 were observed[24]. The engineered strain M0351 was inoculated into newly

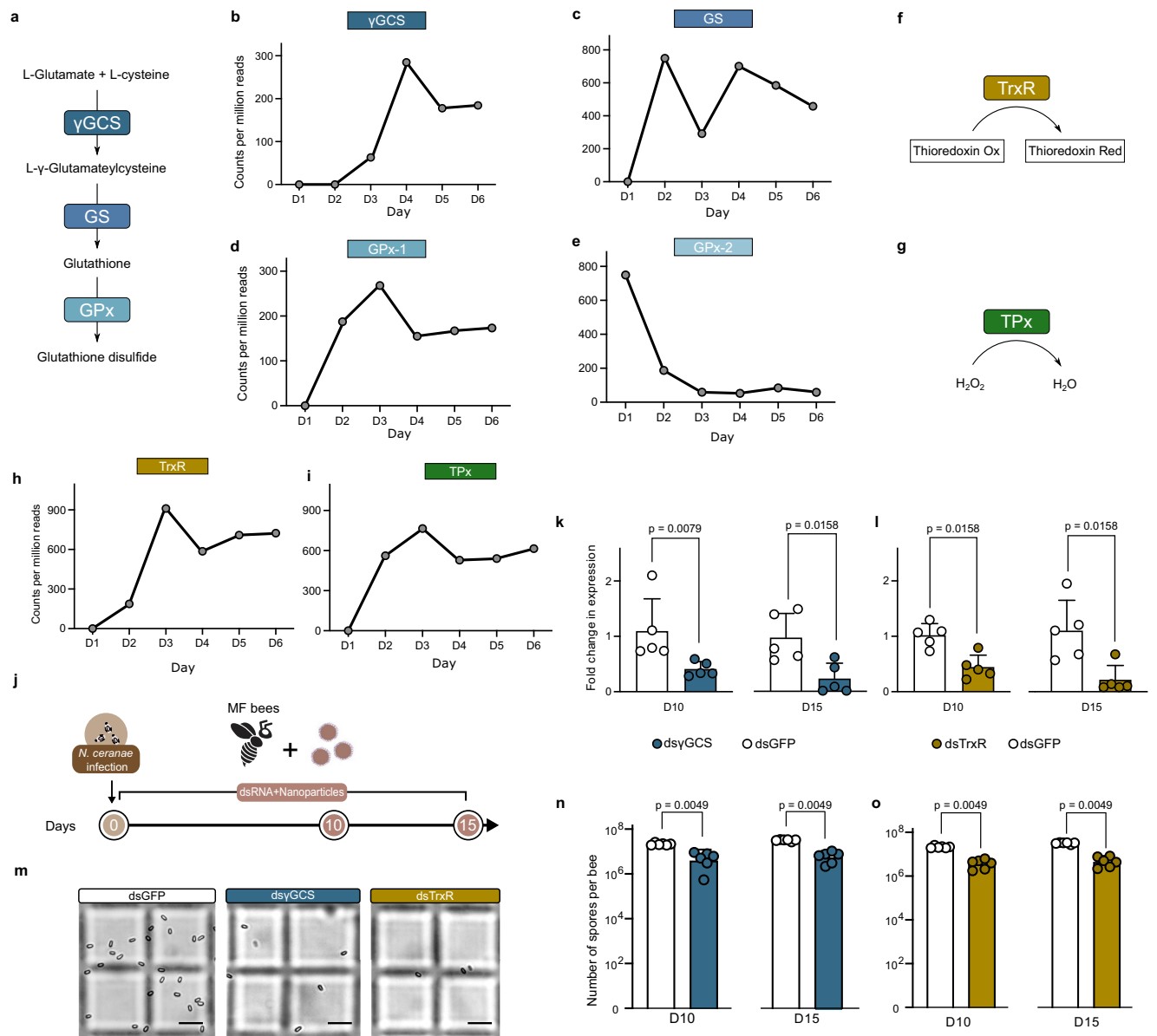

**Fig. 2 | The thioredoxin and glutathione systems of *N. ceranae* are significant for the proliferation in the epithelial cells. a** The glutathione system forms glutathione by γ-glutamyl-cysteine (γGCS), glutathione synthetases (GS) and glutathione peroxidases (GPx) in *N. ceranae*. **b–e** The expression level of the *γGCS, GS, GPx*-1, and *GPx*-2 genes over the infection process. We replotted the mRNA count of *N. ceranae* depending on previous quantitation by Huang et al.[34] **f, g** *N. ceranae* possesses a complete thioredoxin system consisting of thioredoxin reductase (TrxR) (**f**) and thioredoxin peroxidase (TPx) (**g**). **h, i** The expression level of the *TrxR* (**h**) and *TPx* (**i**) genes over the infection process. **j** Knockdown of *N. ceranae γGCS* and *TrxR* gene expression by feeding nanoparticle-mediated dsRNA. **k, l** Relative expressions of the *γGCS* and *TrxR* genes of *N. ceranae* before and after RNAi on days 10 and 15 ($n = 5$ bees for both groups). **m** The load of *N. ceranae* spores was quantified by microscopy using a hemocytometer. **n, o** Silencing of *γGCS* and *TrxR* genes inhibited *Nosema* infection levels ($n = 6$ bees for both groups). Scale bars = 0.025 mm. Statistical analysis was performed by the two-sided Mann–Whitney *U* test. Error bars represent mean SD. Source data are provided as a Source Data file.

emerged bees treated with ampicillin to eliminate native microbiota (Fig. 3a)[25]. We inoculated bee individuals with ~$10^5$ colony-forming units (CFUs) of *GFP*-tagged *S. alvi*. They grew to ~$8.0 \times 10^7$ CFU/bee after 5 days of colonization and persisted stably throughout the 15-day experiments (Fig. 3b). The majority of engineered *S. alvi* cells (~80%) remained functional with a high density of fluorescent signal, while some bacterial cells lost the fluorescence in the guts at the endpoint (day 15; Fig. 3c). While *Snodgrassella* preferentially colonizes the ileum, they also distribute in all compartments of the bee gut[38]. The confocal microscopy showed that 15 days after colonization, the engineered M0351 effectively colonized both the ileum and midguts of 10 inspected bees, showing the same spatial distributions as the wild-type strain (Fig. 3d)[39]. Thus, our results showed that the engineered *S. alvi*

could persistently colonize the honeybee ileum and midgut, and the plasmid pBTK501 functioned reliably in strain M0351 throughout the experiments.

We have shown that *S. alvi*-treated honeybees prevent *N. ceranae* infection by triggering ROS production (Fig. 1), and *N. ceranae* employed the thioredoxin and glutathione system to relieve the intense oxidative stress (Fig. 2). Thus, we engineered *S. alvi* M0351 using plasmid to express dsRNA targeting the glutathione and thioredoxin systems of *N. ceranae*. Target sequences from the *γGCS, GS, GPx* −1, *GPx*−2, *TrxR*, and *TPx* genes were designed and amplified from the cDNA of *N. ceranae* (Supplementary Figs. 4 and 5). Using the Bee Microbiome Toolkit, we assembled plasmids with an inverted arrangement of two promoters (pBTK150, pBTK151) and other

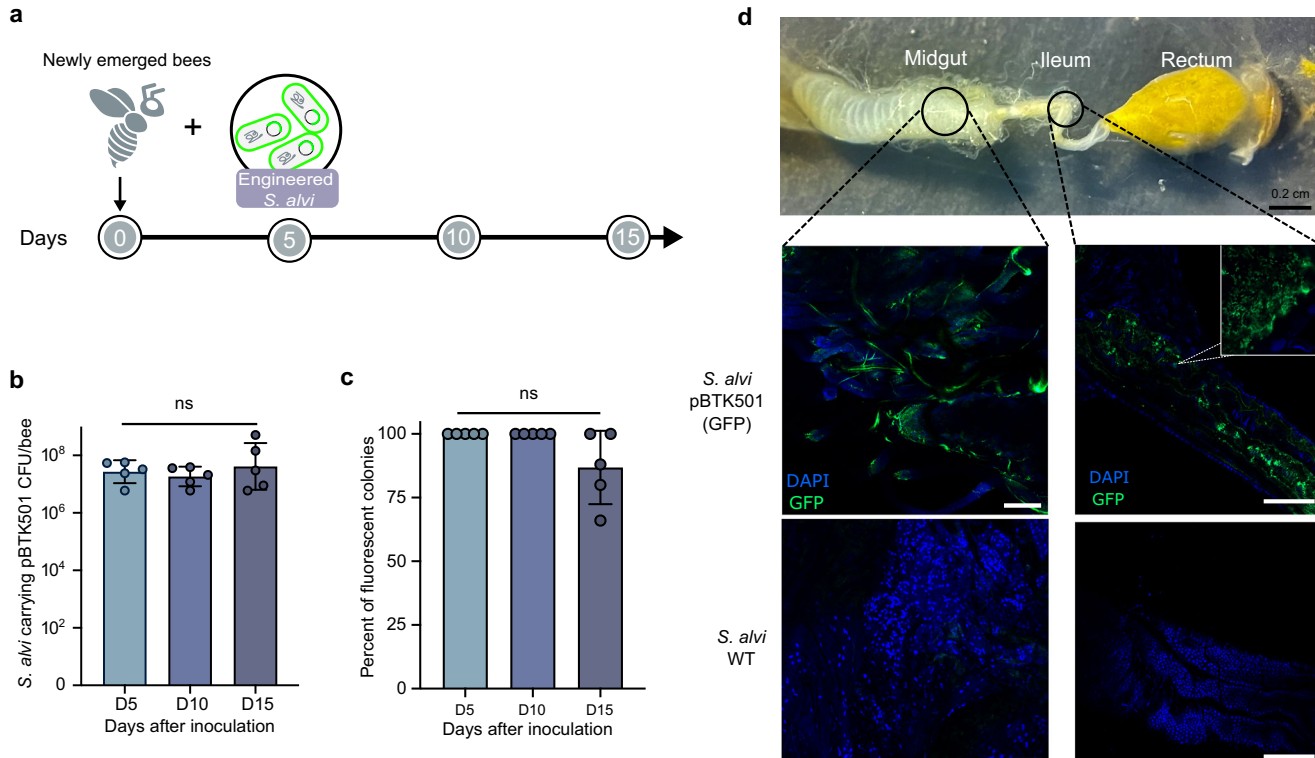

**Fig. 3 | The engineered *S. alvi* M0351 showed stable colonization and function in bee guts. a** Newly emerged bees were colonized with *S. alvi* transformed with a plasmid expressing a green fluorescent protein. The colonization level was checked on days 5, 10, and 15. **b**, **c** Engineered *S. alvi* M0351 stably colonized and expressed GFP continuously over time. Each dot represents an individual bee sample (*n* = 5). Colony-forming unit (CFU). There was no statistical significance (ns) between each

group detected by one-sided ANOVA and Tukey HSD test. Error bars represent mean SD. **d** Engineered *S. alvi* M0351 (Green) colonized both the midgut and ileum of bees. Honeybee gut stained using 4′,6-diamidino-2-phenylindole (DAPI) (blue). For each inoculation condition, three individual bees were inspected during each of the three independent colonization trials independently. Representative images are shown. Scale bars = 200 μm. Source data are provided as a Source Data file.

previously designed parts to produce dsRNA[24,40]. We built six complete dsRNA-producing plasmids targeting different genes and transformed these plasmids into *S. alvi* M0351 by conjugation. We inoculated newly emerged honeybees with ~10^5 cells of *S. alvi* bearing different plasmids that expressed dsRNA corresponding to the GFP coding sequence (pDS-GFP) or those expressed target sequences. Then, the bees were challenged by oral feeding with *N. ceranae* spores (10^4 spores/bee), and 10 days later, we tested whether the *Snodgrassella*-produced dsRNA could inhibit the proliferation of *N. ceranae* (Fig. 4a).

We first extracted the RNA of *N. ceranae* to confirm the depression of targeted pathways. Compared to the pDS-GFP off-target control, a significantly lower expression of target genes from *N. ceranae* was identified (Fig. 4b–g). Expression of all targeted genes is decreased by 50–86% in *N. ceranae* with different recombinant strains, suggesting that the dsRNA is delivered from the engineered *S. alvi* to allow diffusion to the parasite. After 10 days of dsRNA silencing, we evaluated the inhibitory capacity of various engineered *S. alvi* strains by two approaches: spore counting with microscopic observation (Fig. 4h) and qPCR (Fig. 4i). First, both the wild-type *S. alvi* and the pDS-GFP provided protection compared with the controls without symbiont inoculation, confirming the role of *S. alvi* in defending against *N. ceranae*. Engineered *S. alvi* strains expressing γGCS, GPx-1, GPx-2, TrxR, or TPx dsRNA decreased the *Nosema* load when evaluated by spore counts, while they did not show significant difference with bees colonized by wild-type *S. alvi* using qPCR test. However, pDS-GS targeting the GS of the glutathione system showed substantial inhibition of the microsporidia spore invasion evaluating with both approaches. Moreover, we also assessed the effect of mixing engineered strains, and the inhibitory effect by a mixture

of bacteria delivering all six dsRNA was better than most colonization by single strains. But the effect of mixed strains was similar to that of the pDS-GS.

## Discussion

*N. ceranae* is a microsporidian parasite initially identified from the *A. cerana* in the 1990s[41] and later detected in different honeybee species worldwide, becoming a globally distributed pathogen[30]. In this study, we found that *N. ceranae* employs the thioredoxin and glutathione system to relieve oxidative stress from the host for the adaptation in the midgut epithelium (Fig. 5). We showed that the core gut bacteria, *S. alvi*, triggers the redox response involved in honeybee gut immunity, which inhibits the proliferation of *N. ceranae* by up to 85.5%. Moreover, we successfully constructed engineered *S. alvi* M0351 based on the Bee Microbiome Toolkit and the Functional Genomics Using Engineered Symbionts procedure (FUGUES) to continuously produce dsRNA for critical genes of the *N. ceranae* thioredoxin and glutathione systems. Engineered *S. alvi* can stably re-colonize bees and repress the parasite's thioredoxin and glutathione system-related gene expression. The *GS* of the glutathione system is a relatively more effective targeted gene for *N. ceranae* inhibition. Furthermore, the inhibitory effect by the mixture of all six engineered *S. alvi* strains also showed a higher inhibition than many single strains but similar to pDS-GS.

The microsporidia *N. ceranae* is an obligate intracellular parasite that develops in the ventricular epithelia of *A. mellifera*, and the spores can spread the infection quickly across epithelial cells[42]. The concept that the immune activation of honeybees plays a role against *N. ceranae* is not new, but the exact mechanism remains unclear. In all invertebrates, including insects, the adaptive system is missing, and hence, defense is entirely ensured by the innate immune

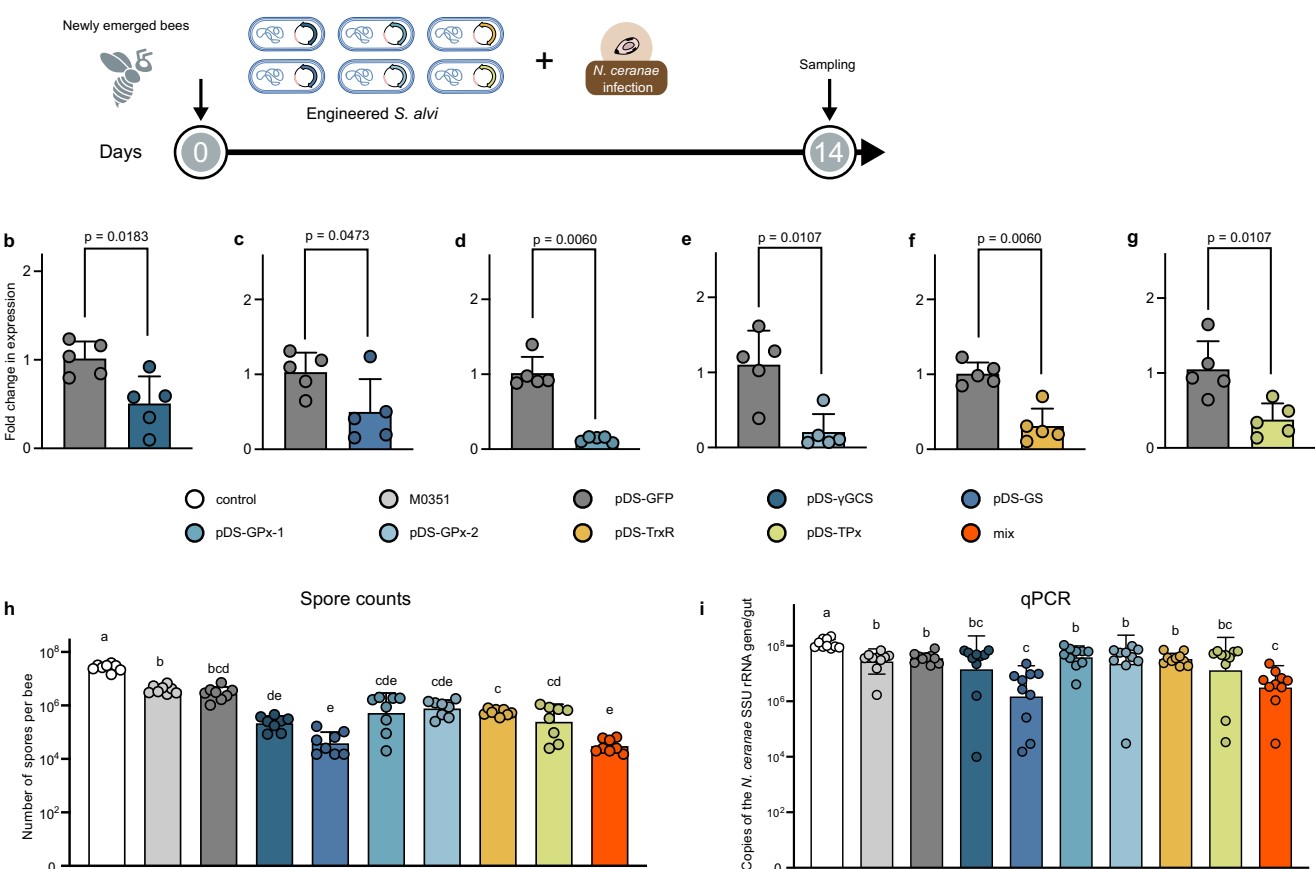

**Fig. 4 | Recombinant *S. alvi* M0351 strains engineered to deliver dsRNA inhibit the infection of *N. ceranae*. a** Newly emerged bees were colonized with engineered *S. alvi* and orally infected with *N. ceranae*. **b–g** Engineered *S. alvi* strains to produce dsRNA targeting genes of the thioredoxin and glutathione systems inhibited the gene expression of *N. ceranae* in the midgut (*n* = 5). Two-sided Mann–Whitney *U* test. **h**, **i** Inhibition of *N. ceranae* infection by engineered *S. alvi*

M0351 strains. *Nosema* infection was assessed by both spore counts (**h**) (*n* = 8 bees for all groups) and qPCR (**i**) (*n* = 10 bees for all groups). The overall significance was first assessed by one-sided ANOVA (*p* < 0.01). Brown–Forsythe and Welch test was used for multiple comparisons, the letters above each bar stand for statistical differences among the treatments (*p* < 0.05). Error bars represent mean SD. Source data are provided as a Source Data file.

system. The innate immune system provides organisms with a rapid, non-specific first line of defense against colonization by pathogenic microorganisms[43]. Honeybees' cell-mediated innate immune system consists of hemocytes, which produce ROS essential for cell signaling and pathogen clearance[44]. ROS are produced during the recognition and phagocytosis response against the foreign bodies inside the cells[45]. In insects, ROS was essential in the fight against parasites and pathogenic bacteria in the gut[30,46]. The production of ROS has been observed following the infection of *Aedes aegypti* mosquito by different microsporidian species[47]. In honeybees, the ROS genes are overexpressed in the midgut upon the spore infection, suggesting that the increased ROS level is an immune response against intracellular parasitism[19]. Our results confirmed that the ROS-associated immune response in the epithelial cells is indispensable for preventing microsporidia infection in bees.

Two ROS-producing enzymes, the NADPH oxidases Duox and Nox, are identified in *A. mellifera*, similar to *Drosophila*. Here, we found that the expression of both *Duox* and *Nox* genes was activated in the midgut of bees post-colonization by *S. alvi*. It has been documented that *S. alvi* protects honeybees from the opportunistic pathogen *Serratia marcescens* by triggering the Toll and Imd pathway to upregulate the expression of host AMPs abaecin, apidaecin, and hymenoptaecin[48]. *S. alvi* colonizes the honeybee gut in contact with the gut epithelia and forms a dense biofilm, which may stimulate pattern recognition

receptors such as Toll-like receptors of bees[48]. Interestingly, the activation of Toll-like receptors possibly conjugates the NADPH oxidases, which are involved in the ROS production on the membranes of the endosome of cells[49]. Notably, bees mono-colonized with *S. alvi* strain inhibit the proliferation of *N. ceranae* by up to 85.5%. In contrast, bees colonized by other gut members did not show a significant reduction of *N. ceranae*. Thus, the ROS immune response may be activated by *S. alvi* by regulating the Toll signaling and the NADPH complex. The ROS system is also a gut-immune immune response involved in gut-microbe homeostasis[18]. In honeybees, the *N. ceranae* infection can perturb the gut composition, and a normal microbiota is required for host resistance to *N. ceranae*[50,51]. The core gut members of honeybees, including the *S. alvi* and *L. apis*, have been shown to activate the humoral innate immune system to produce AMPs, which protect against pathogens[22,23]. Our data show that the bee gut bacteria also play a pivotal role in controlling microsporidia invasion by inducing de novo generation of ROS[18].

To protect itself against host defenses of ROS, the parasite may have an internal antioxidant system to maintain a normal redox state. For example, *Plasmodium falciparum* scavenges ROS from hosts by employing antioxidant systems, including the NADPH-dependent thioredoxin and the glutathione system[37,52]. We found that *N. ceranae* also possesses balanced redox pathways, including the thioredoxin and glutathione systems, which may be necessary for counteracting

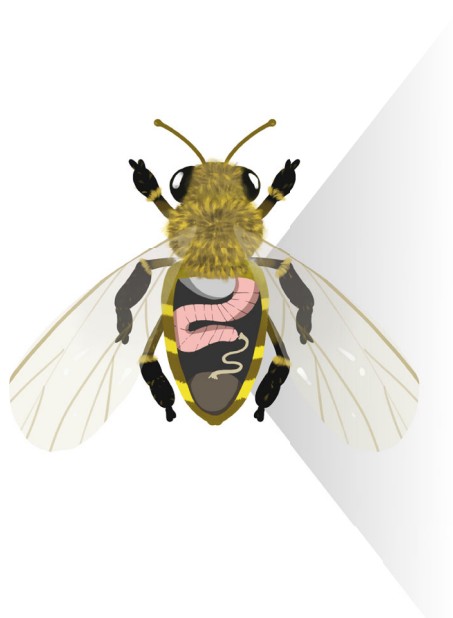

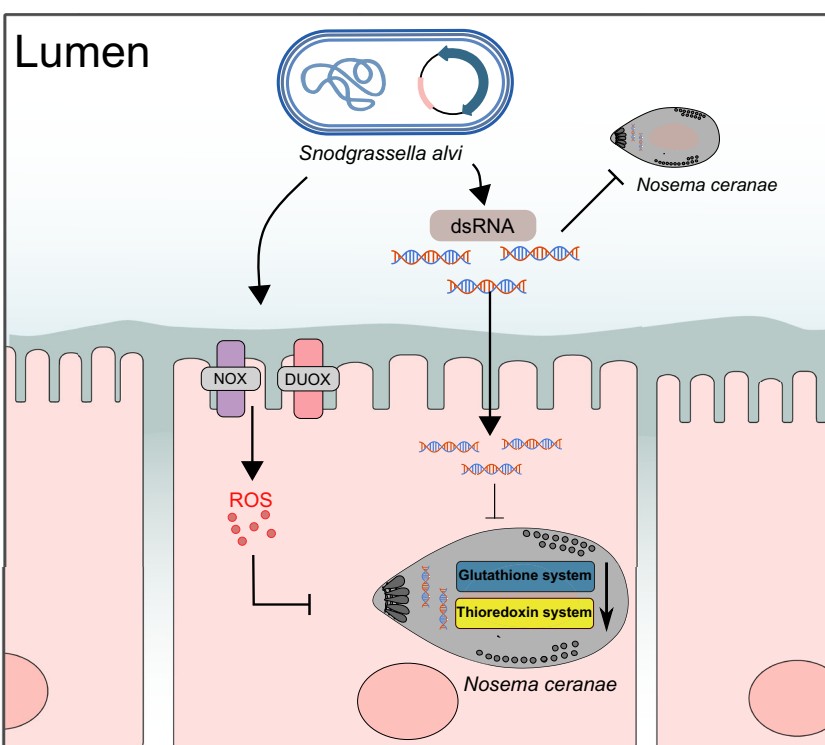

**Fig. 5 | Graphical summary of the *N. ceranae* inhibition by the engineered *S. alvi* M0351 strains.** By triggering *Duox* and *Nox* genes, *S. alvi* induces Reactive oxygen species (ROS) production in the midgut of honeybees, which may kill *N. ceranae* via perturbation of redox homeostasis. *N. ceranae* may employ the thioredoxin and glutathione antioxidant systems to relieve the intense oxidative stress for intracellular proliferation. Engineered *S. alvi* M0351 expressing the double-strand RNA (dsRNA) corresponding to the genes of the oxidation-reduction systems of *N. ceranae* can significantly inhibit parasitism in the honeybee gut.

ROS attacks from the host. Analyzing the time series of gene expression of *N. ceranae* colonizing the bee gut, we found significant upregulation in the expression of thioredoxin and glutathione system genes, suggesting that the maintenance of the normal redox state is significant in the invasion process of *N. ceranae*. Previous studies that used RNAi targeting on variable *N. ceranae* genes, such as the ptp3 and the SWP, reduce parasite load and improve the physiological performance of honeybees[27,53]. Our results showed that the delivery of dsRNA corresponding to γGCS and TrxR led to a significant reduction in spore load by 82% and 85%, respectively, indicating that the endogenous antioxidant enzymes of *N. ceranae* provide a novel therapeutic target for the control of the parasitic invasion in honeybees.

Although RNAi is widely used, it is expensive for application in agricultural fields, and its efficiency is also unsatisfactory. Recently, researchers have engineered host-associated bacterium to produce dsRNA as a novel delivery modality. Taracena et al. used engineered *Escherichia coli* to produce dsRNA to control the *Rhodnius prolixus* parasite, a vector of the Chagas disease[54]. Although laboratory *E. coli* is easy to be engineered, the symbiotic bacteria with minimal fitness cost and stable association with the host are more promising for in vivo treatment[55]. Leonard et al. have recently designed the Functional Genomics Using Engineered Symbionts (FUGUES) procedure to engineer the native bacterial species of honeybees[25]. The engineered *S. alvi* produces dsRNA to inhibit parasitic *Varroa* by inducing mite RNAi response[40]. It documents that the RNA produced by the recombinant strains can be transported to the gut, hemolymph, and head of bees, which alters host physiology and behavior. *N. ceranae* infects and proliferates intracellularly in the epithelial cells of the midgut[41]. While *Snodgrassella* mainly localizes to the ileum region of the hindgut, it can also be found in the midgut[38]. Our results illustrated that the recombinant *S. alvi* strain stably re-colonizes both ileum and midgut post-inoculation, and the plasmid exhibits a robust expression in vivo. Thus, the dsRNA produced by the symbiotic *S. alvi* may enter the midgut

cells and destroy the redox homeostasis of *N. ceranae*. Here, we targeted all six genes involved in the glutathione and thioredoxin systems of the microsporidia. Interestingly, the repression of the *GS* gene showed the highest inhibition efficiency. GS is not subject to feedback inhibition by GSH and is important in determining overall GSH synthetic capacity, specifically under pathological conditions[56]. In addition, the inhibition of a mixture of the recombinant strains is more significant than the single-targeting strains except for pDS-GS, implying the magnitude of knockdown with the mix might be primarily driven by pDS-GS or potential synergistic activity. *S. alvi* and other bee gut bacteria can be naturally transmitted within the colony via social contact, and engineered *S. alvi* strains are transferred between co-housed bees, which suggests that native gut bacteria can facilitate the treatment of individual bees from an entire colony[40]. Although gene escape is a major ethical issue for the application of engineered bacteria, honeybee symbiotic bacteria are generally restricted to the bee gut environment[21]. Moreover, *S. alvi* shows even more strict host specificity that only specific strains are associated with different bee species or even individuals within the colony[57]. Alternatively, engineering auxotrophies may also provide a containment approach to reduce the risks associated with bacterial release[58]. Thus, symbiont-mediated RNAi delivers a new tool to improve resilience against current and future challenges to honeybee health.

## Methods

### Generation of microbiota-free and mono-colonized honeybees
Insects: honeybees (*A. mellifera*) used in this study were collected from colonies kept in the experimental apiary of the China Agricultural University. Pupae and newly emerged bees used in all the experiments were obtained from brood frames taken from the experimental hives and kept in an incubator (Bluepard, Shanghai, China) at 35 °C, with a humidity of 50%. There is no current requirement regarding insect care and use in research. Honeybees were cared for daily with adequate

food during the experimental period. For tissue collection, bees were collected gently and immediately euthanized by $CO_2$ anesthesia before dissection to reduce any unnecessary duress.

All honeybee gut bacterial strains used in this study are listed in Supplementary Table 1. *S. alvi* M0351, *B. choladohabitans* W8113, and *G. apicola* B14384H2 isolated from honeybee guts were grown on Heart Infusion Agar (Oxiod, Hampshire, UK) supplemented with 5% (vol/vol) sterile sheep blood (Solarbio, Beijing, China) at 35 °C under a $CO_2$-enriched atmosphere (5%). *B. mellis* W8089 and *L. apis* W8172 were grown on De Man, Rogosa and Sharpe agar plates (Solarbio) supplemented with 0.1% L-cystine and 2.0% fructose at 35 °C under a $CO_2$-enriched atmosphere (5%).

MF bees were obtained as described by Zheng et al.[59] In brief, we manually removed pupae from brood frames and placed them in sterile 500-ml plastic bins. Newly emerged MF bees were kept in axenic cup cages with sterile sucrose syrup for 24 h. For each mono-colonization setup, 20–25 MF bees were placed in an axenic 500-ml cup cage and fed bacterial culture solutions for 24 h. Colonization levels were determined by CFUs from dissected guts, as described by Kwong et al.[59] 1 ml of sterilized 1× phosphate-buffered saline (PBS) (Solarbio) was combined with 1 ml of sterilized sucrose solution (50%, wt/vol) and 0.3 g of sterilized pollen for the MF group. For the mono-colonization bees, glycerol stock of bee gut strains was resuspended in 1 ml of sterilized 1× PBS at a final concentration of ~$10^8$ CFUs/ml (determined by counting colonies on plates) and then mixed with 1 ml of sterilized sucrose solution (50%, wt/vol) with 0.3 g of sterilized pollen. The bees were incubated at 35 °C and RH 50% until day 7.

### N. ceranae spore purification

*N. ceranae* spores were isolated from worker honeybees collected from heavily infected colonies in the summer of 2022. After immobilizing bees by chilling them on ice, the guts were removed from individual bees with forceps. The midguts of infected honeybees were homogenized in distilled water and filtered using NO. 4 Whatman filtering paper. The filtered suspension was centrifuged at 3000 × g for 5 min, and the supernatant was discarded. The resuspended pellet was purified on a discontinuous Percoll (Sigma-Aldrich, St. Louis, MO, USA) gradient of 5 ml each of 25%, 50%, 75%, and 100% Percoll solution. The spore suspension was overlaid onto the gradient and centrifuged at 8000 × g for 10 min at 4 °C using a Sigma 1–14 K centrifuge (Sigma-Aldrich). The supernatant was discarded, and the spore pellet was washed by centrifugation and suspension in distilled sterile water[60]. The number of spores was quantified using a Fuchs−Rosenthal hemocytometer (Blaubrand, Wertheim, Germany). The identity of the isolated *N. ceranae* or *N. apis* was determined by amplifying the ribosomal RNA gene sequences with species-specific primers (Supplementary Table 3)[61].

### Bees mono-colonized with gut symbionts challenged with N. ceranae

To accurately control the number of *N. ceranae* cells infecting each bee individual, bees were orally fed the same amount of *N. ceranae* spores. Each bee was starved for 2 h and given 2 µl of a sterilized sucrose solution (50%, wt/vol) containing $10^4$ *N. ceranae* spores. After 10 days, the number of spores in the intestinal specimen of infected bees was quantified as described by Huang et al.[50] The midguts were dissected, resuspended in 500 µl of double-distilled water, and then subjected to vortex mixing. The suspension was put onto the Fuchs−Rosenthal hemocytometer (Blaubrand) for microscopic observation (Nikon, Tokyo, Japan).

### Honeybee gut RNA extraction and quantitative PCR

Each dissected gut was homogenized with a plastic pestle, and total RNA was extracted from individual samples using the Quick-RNA MiniPrep kit (Zymo, Irvine, CA, USA). RNA was eluted into 50 µl of RNase-free water and stored at −80 °C prior to reverse transcription. cDNA was synthesized using the HiScript III All-in-one RT SuperMix Perfect for qPCR (Vazyme Biotech, Nanjing, China). Quantitative real-time PCR was performed using the ChamQ Universal SYBR qPCR Master Mix (Vazyme Biotech) and QuantStudio 1 Real-Time PCR Instrument (Thermo Fisher Scientific, Waltham, MA, USA) in a standard 96-well block (20-µl reactions; incubation at 95 °C for 3 min, 40 cycles of denaturation at 95 °C for 10 s, annealing/extension at 60 °C for 20 s). The primers for the genes of *Duox* (LOC551970) and *Nox* (LOC408451) of *A. mellifera* were designed with IDT qPCR PrimerQuest Tool (https://www.idtdna.com/pages/tools/primerquest) (Supplementary Table 3). The *actin* gene of *A. mellifera* was used as the control, and the relative expression was calculated using the $2^{-\Delta\Delta CT}$ method[62]. No data were excluded from the analyses.

### In vivo detection of reactive oxygen species

Three days after inoculation, the midguts of the honeybees mono-colonized with *S. alvi* M0351 and the MF bees were dissected in PBS containing 50 µM intracellular ROS-sensitive fluorescent dye dihydroethidium (Thermo Fisher Scientific). The tubes were placed in the dark for 10 min at room temperature. Then, the midguts were washed twice with a fresh dye-free PBS, and the tissues were immediately transferred to µ-Dish$^{35\ mm,\ high}$ microscope dishes (Ibidi, Martinsried, Germany). We imaged the gut tissues on a Zeiss 910 Laser Scanning Confocal microscope with a 20× objective (Carl Zeiss Microscopy GmbH, Jena, Germany).

### Measurement of the H₂O₂ production

The generation of $H_2O_2$ was determined using the Hydrogen Peroxide Assay Kit (Beyotime Biotech, Shanghai, China). In this assay, $H_2O_2$ converts $Fe^{2+}$ to $Fe^{3+}$, which then reacts with xylenol orange dye to become purple with a maximum absorbance at 560 nm. The midguts were homogenized in 200 µl lysis buffer and centrifuged at 12,000 × g at 4 °C for 5 min, and the supernatant was collected. Aliquots of 50 µl of supernatants and 100 µl of test solutions from the Hydrogen Peroxide Assay Kit were incubated at room temperature for 20 min and measured immediately with a spectrometer at 560 nm. The measurement was repeated three times for each sample. No data were excluded from the analyses.

### RNA isolation of N. ceranae

To extract the RNA of *N. ceranae*, the honeybee gut was individually transferred into 2 ml tubes. Each tube contained 100 µl sterile 1.4-mm zirconium silicate grinding beads (Quackenbush, Crystal Lake, USA). One milliliter of TRIzol reagent (Thermo Fisher Scientific) was added to the tube, disrupting the samples using the FastPrep. The samples were treated with DNase I (Thermo Fisher Scientific) to remove genomic DNA contamination. The purity and quantity of RNA samples were determined using a NanoDrop 8000 spectrophotometer (Thermo Fisher Scientific). cDNA was synthesized using the HiScript III All-in-one RT SuperMix Perfect for qPCR (Vazyme Biotech) and stored at −20 °C.

### Nosema inoculation and nanocarrier-mediated dsRNA feeding assay

To produce the double-stranded RNA of the *γGCS* (AAJ76_1100057370) and TrxR (AAJ76_5800012528) genes, the coding regions of the genes were amplified from *N. ceranae* cDNA with forward and reverse primers containing the T7 promoter sequence at their 5′ends (5′-TAATACGACTCACTATAGGGCGA-3′). The partially amplified segments of the genes were cloned into the pCE2-TA-Blunt-Zero vector (Vazyme Biotech) and verified by Sanger sequencing. The fragment was amplified from the plasmid using specific primers with a T7 promoter and then used for dsRNA synthesis using the T7 RNAi Transcription Kit (Vazyme Biotech). The fragment amplified from the *GFP* gene

(MH423581) was used as the control. The sequences of the primers are given in Supplementary Table 3. Here, we used the star polycation as a gene nanocarrier to protect dsRNA molecules from enzymatic degradation and promote their translocation across cell membranes[63]. The nanocarrier was gently mixed with γGCS and TrxR dsRNA at a mass ratio of 1:1. (The final concentration for both SPc and dsRNA was 100 ng/μl.) The final concentrations for dsRNA + nanocarrier and sucrose were 100 ng/μl and 50% (wt/vol), respectively. MF bees were kept without food for at least 2 h before the subsequent *N. ceranae* inoculation. Individual bees were fed 2 μl of spores suspensions prepared by mixing purified spores into sterilized sucrose solution (50%, wt/vol) (~10⁴ spores/μl). From the day after *N. ceranae* inoculation, honeybees from each treatment were fed on different dsRNA mixtures in an incubator at 35 °C The dsRNA mixture was supplied daily, and each bee ingested about 10 μg of dsRNA per day.

The treatment effect of dsRNA was determined by comparing the spore production rate for individual honeybees. The *N. ceranae* spore production rate was measured by counting the spores from the extracted midgut of live honeybees 15 days after inoculation. To investigate the effect of dsRNA treatment on the expression of each target gene of *Nosema*, qRT-PCR was performed after 15 days of dsRNA treatment. After extracting the midguts from honeybees treated with *Nosema* and dsRNA, the total RNA was extracted. cDNA was synthesized using the HiScript III All-in-one RT SuperMix Perfect for qPCR (Vazyme Biotech). Each gene-specific primer is given in Supplementary Table 3. The β-tubulin gene of the *N. ceranae* was used as the control, and relative expression was analyzed using the $2^{-\Delta\Delta CT}$ method[62]. No data were excluded from the analyses.

## Vector construction to express dsRNA expression and *S. alvi* M0351 engineering

All the plasmids (pYTK002, pBTK150, pBTK151, pYTK072, pBTK301, pBTK401) and *E. coli* MFD*pir*[25] were kindly donated by the Moran Lab and Barrick Lab (University of Texas at Austin). These plasmids belong to the bee microbiome toolkit, which is broad-host-range plasmids built by the RSF1010 replicon. We designed dsRNA-producing plasmid parts based on the previously published Bee Microbiome Toolkit and functional genomics using engineered symbionts procedure (FUGUES) (Supplementary Fig. 4)[25]. First, PCR is used to amplify the knockdown region γGCS, GS, GPx-1, GPx-2, TrxR, and TPx from the cDNA of *N. ceranae* and append BsaI cut sites to each end. Following PCR, amplicons are purified and cloned into a dsRNA expression vector. We combined previously designed parts pYTK002 (Type 1), pBTK150(Type 2), pBTK151(Type 4), pYTK072 (Type 5), pBTK301 (Type 6–7), and pBTK401 (Type 8) (Addgene_65109, Addgene_183127, Addgene_65179, Addgene_183126, Addgene_110593, Addgene_110597), and dsRNA target sequence (Type 3) to assemble complete plasmids that express dsRNA of the target sequence[40]. Golden Gate assembly reactions were performed as previously described[24], and enzyme BsaI-HFv2 (New England Biolabs, Beverly, MA, USA) was used to increase assembly efficiency.

Assemblies were transformed into electroporated into *E. coli* donor strain MFD*pir*, which is a diaminopimelic acid (DAP) auxotroph mutant[64]. The transformed cells were then screened on LB agar plates containing 0.30 mM DAP and 100 μg/ml ampicillin. The conjugation of these plasmids into bee gut bacteria was performed using previously described methods[24]. Briefly, the conjugation process involved mixing overnight cultures of donor and recipient bacteria in roughly equal proportions based on optical density. These mixtures were then incubated overnight on a non-selective agar plate supplemented with DAP. On the following day, the conjugation mixture was resuspended in PBS and plated on selective media without DAP but containing ampicillin (100 μg/ml) in dilutions. After obtaining antibiotic-resistant colonies, we confirmed stable transformation by passaging them again on selective media. These transconjugants were confirmed to be pure

*S. alvi* cultures by performing 16S rRNA sequencing to ensure no unexpected contaminants had been introduced during the conjugation process.

We scraped the engineered *S. alvi* grown on the plates into PBS. These cells were spun in a centrifuge (3824 × g, 5 min) and resuspended in 500 μl PBS. Engineered *S. alvi* was diluted in 500 μl sterilized 1× PBS at a final concentration of ~10⁸ CFUs/ml and combined with 500 μl of a 1:1 sucrose (100%, wt/vol): water solution supplemented with 200 μg/ml ampicillin. We fed engineered *S. alvi* solutions to age-controlled newly emerged worker bees for 24 h (pDS-γGCS, pDS-GS, pDS-GPx-1, pDS-GPx-2, pDS-TrxR, pDS-TPx) and non-targeted (pDS-GFP) served as a negative control group. The next day, each bee was given 2 μl of a sterilized sucrose solution (50%, wt/vol) containing 10⁴ *N. ceranae* spores. After 10 days, honeybee gut was collected to quantify the number of *N. ceranae* spores, and gene knockdown was validated using qPCR on the cDNA of *N. ceranae* synthesized as described above.

To test whether engineered *S. alvi* robustly colonizes bees, we inoculated bees with *S. alvi* transformed with a plasmid expressing GFP. Firstly, we transformed strain M0351 with a stable plasmid pBTK501 expressing GFP from the Bee Microbiome Toolkit platform (Addgene_110602)[24] and inoculated bees with *S. alvi* M0351::pBTK501 (~10⁵ CFU/bee). After every 5 days, we dissected bees, homogenized their whole guts in 500 μl PBS, and plated dilutions onto Heart Infusion Agar plates with a final concentration of 100 μg/ml ampicillin to estimate CFUs of *S. alvi* in the gut. The number of fluorescent and non-fluorescent colonies on the plates was quantified to track the stability of engineered strains over time. After 15 days, we immobilized bees with $CO_2$, placed them on ice, and then dissected out whole guts. We put the guts on μ-Dish³⁵ ᵐᵐ, ʰⁱᵍʰ microscope dishes (Ibidi) and added 1 μL of PBS to prevent gut tissue from drying out. Whole guts were imaged using a Leica SP8 Laser Confocal Microscope with a 20× objective. The images were collected by the LAS X software. To determine the colonization site of the engineered *S. alvi* strain, we used DAPI and GFP fluorescent markers with excitation wavelengths of 360 and 488 nm and emission wavelengths of 460 and 511 nm, respectively. We used 700–720 of Smart Gain for both channels, as suggested by the user manual.

## Genomic DNA extraction and qPCR for quantification of *N. ceranae*

The guts were grounded individually in liquid nitrogen. The genomic DNA was extracted with a FastPure Blood/Cell/Tissue/Bacteria DNA Isolation Mini Kit (Vazyme Biotech). *N. ceranae* spores were determined by qPCR using the ChamQ Universal SYBR qPCR Master Mix (Vazyme Biotech). *N. ceranae*-specific primer sets are listed in Supplemental Table 3. All qPCRs were performed in 96-well microplates on a QuantStudio 1 real-time PCR system (Thermo Fischer Scientific). Melting curves were generated after each run (95 °C for 15 s, 60 °C for 20 s, and increments of 0.3 °C until reaching 95 °C for 15 s). As previously described[8], Standard curves were generated using serial dilutions of target DNA fragments (i.e., purified PCR products) ranging from 10⁻² to 10⁻⁸. No data were excluded from the analyses. The data were analyzed using the QuantStudio Design and Analysis Software (version 1.5.0; Thermo Fisher Scientific).

## Statistics and reproducibility

Comparison of the *N. ceranae* spore counts between the MF group and the mono-colonization groups, the expression level of the *Duox*, *Nox*, and thioredoxin/glutathione system genes of different groups were tested by two-sided Mann–Whitney *U* test. The $H_2O_2$ concentration in the midgut of MF and mono-colonized bees was tested by multiple two-tailed *t*-tests. Comparison of the colonization levels of engineered *S. alvi* M0351 and wild-type *S. alvi* M0351 was detected by one-sided

ANOVA and Tukey HSD test. *Nosema* infection level (spore counts and qPCR tests) among different groups was detected by one-sided ANOVA and Brown–Forsythe and Welch test.

The exact value of *n* representing the number of groups in the experiments described was indicated in the figure legends. Any additional biological replicates are described within the Methods and the Results. No statistical method was used to predetermine the sample size. No data were excluded from the analyses. The experiments were not randomized. The Investigators were not blinded to allocation during experiments and outcome assessment.

### Reporting summary

Further information on research design is available in the Nature Portfolio Reporting Summary linked to this article.

## Data availability

All data generated in this study are provided within the manuscript files. All vectors generated in this work can be requested by contacting H.Z. (hao.zheng@cau.edu.cn). The raw RNA-seq data of the gene expression file of *Nosema ceranae* are derived from Huang et al.[34] and the source data are available in Table S1 of the original publication. Source Data are provided with this paper.

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

## Acknowledgements

This work was supported by National Key R&D Program of China (Grant No. 2019YFA0906500) and National Natural Science Foundation of China Project 32170495 to H.Z. and National Natural Science Foundation of China Project 31760715 to X.X. We thank the Moran Lab and Barrick Lab (University of Texas at Austin) for donating plasmids. We thank Elijah Powell, Jeffrey Barrick, and Nancy Moran for their suggestions in gut bacteria engineering. We thank Jie Shen and Shuo Yan (China Agricultural University) for their assistance in the nanoparticle RNA delivery system.

## Author contributions

H.Z. supervised the study; H.Z., X.Z., and H.L. designed the study; H.L., Hao W., Haoqing W., and W.Z. collected samples and performed the RNAi experiments; Z.Z., X.X., and J.G. collected and identified the Nosema samples; H.L., X.H., and L.M. generated data and performed the data analyses with contributions from Hao W.; X.H., H.Z., X.Z., and H.L. prepared the manuscript.

## Competing interests

The authors declare no competing interests.
