## [Peer Review File · Nature Communications]

REVIEWER COMMENTS

Reviewer #1 (Remarks to the Author):

In this work, Lang et al provide an important contribution to our understanding of how infection by the bee pathogen *Nosema ceranae* is affected by the bee microbiota and how engineered gut symbionts may be used to protect honey bees from this pathogen. The authors' findings support an important role for reactive oxygen species in honey bee resistance to *Nosema*, and suggest in turn that *Nosema* antioxidant systems are important for successful infection. Through a series of experiments with bees given a controlled microbiota the authors further link production of ROS in bee guts to the presence of *Snodgrassella alvi*, a core bee gut microbiome member. Finally, the authors build on recently published work and adapt a *Snodgrassella* dsRNA expression system, previously used to combat other bee pathogens, to directly target *Nosema* and demonstrate that colonization with these engineered bacteria can lower *Nosema* spore counts. Overall, this work provides useful insight into mechanisms of *Nosema* infection and demonstrates the repeatability and extensibility of the approach of using engineered *S. alvi* to produce dsRNA to protect bee health.

I commend the authors on this interesting and creative multidisciplinary work, and I believe it will be a useful contribution to the field. I do, however, have concerns with how some experiments were carried out and interpreted, and I believe there are several missing pieces of methodological information that are essential to fill in.

Major concerns:

-The findings reported in Fig 1. and discussed in lines 88-91 are over-interpreted compared to the evidence presented. First, the authors provide no evidence that the inoculation with their bacteria were successful. While bee gut bacteria generally colonize microbiota-free (MF) bees well, there is extensive strain to strain variation in colonization. These are newly isolated strains with unknown colonization dynamics. In order to claim a difference between *S. alvi* and the other bacteria, the authors should verify that these strains of bacteria can recolonize bees, or the observed differences could be due to lack of colonization. The authors can do this by reporting CFU counts from the guts of a sample of colonized bees compared to MF bees prior to inoculation. Second, 3 bees per group is an unacceptably low sample size to draw robust conclusions for Fig 1B. As the authors do not report cupcage configuration for this experiment, I assume that each point is a single bee from the same cup. I encourage the authors to repeat the experiment shown in Fig 1B with a larger sample size (≥ 5 bees from ≥ 2 independent cup cages per condition, or similar) to verify the findings.

- In lines 96-103 the authors appear to be making the connection between upregulation of *Duox* and *Nox* uniquely by *S. alvi* as contributing to its ability to lower *N. ceranae* infection. This interpretation

would be strengthened if inoculation by other bacteria did not also trigger Duox and Nox during colonization. However, no data is shown on the possible upregulation of Duox and Nox by the other gut bacteria initially tested. I believe the authors could perform qPCR on bees colonized with the other bacteria for Duox and Nox, or explicitly mention that this upregulation of Duox and Nox may not be unique to *S. alvi*.

-The findings shown in Figure 2A-G and discussed in lines 124-137 require more methodological explanation in the main text and in the methods. It is unclear if the authors performed a new analysis that involved remapping and quantifying the previously published RNA-seq reads, or if the authors relied on the previous quantitation and plotted charts for their genes. If the former, the methods section should be updated to include all the computation steps involved (mapping, counting, etc). If the latter, the methods should still be clarified and relevant caveats with the original data, such as no biological replication, be specified.

-Please provide more methodological details related to the microscopy shown in Figure 3D and discussed in 179-181. Include how the gut sections were prepared and whether they were fixed or imaged without fixation. Include the microscope, objective, and software used for collection of the images, as well as the wavelengths used for each channel and the exposure time for each image. Discuss if any adjustments were made to brightness for display. I am especially surprised to see the robust colonization of fluorescent *S. alvi* in the midgut, as there is usually substantial autofluorescence in the midgut that makes such visualization difficult. The images in Fig3D are also quite small and difficult to inspect closely- perhaps the authors could include full size images in the supplement or deposit raw images with an appropriate data repository.

-Figure 4H, Lines 223-226, 253-254, and 344-350. I commend the authors for using a mixture of strains to deliver multiple dsRNAs, and it has a strong effect. However, repeatedly the authors claim that “inhibitory effect by a mixture of bacteria delivering all six dsRNA was better than the colonization with single strains” or similar. But the data in Fig 4H shows no significant difference between the mix and pDS-GS individually. Thus, the magnitude of knockdown seen with the mix could be primarily driven pDS-GS instead of some synergistic activity between them. This should be made clear in the text, and relevant lines changed indicate the mix did better than many, but not all, individual strains.

-Line 293-295. As discussed above, I don't think the current evidence is strong enough to conclude this reduction is specific to *S. alvi* compared to other bacterial members of the gut. Either include additional replication or soften language that claims this is unique to *S. alvi*.

-I have some concern that the only assay used to assess Nosema infection throughout is spore counts. Was there any effect on bee survival in these experiments? It could also be appropriate to utilize qPCR to assess Nosema burden for some experiments, as described by the COLOSS Bee Book methods on Nosema. Including an additional unbiased estimator of Nosema burden would strengthen the claims made regarding the experiments in Figure 4.

Minor concerns:

- In the introduction the authors should clearly introduce and reference previous efforts to use dsRNA to target Nosema, as there are many previous reports of this approach (See work by J. Evans and others). This need not take away the novelty of what they do, but is important context and helps justify why using dsRNA with *S. alvi* is reasonable

- Please provide more methodological details on the provenance of the bacterial strains in this work. How did you verify the species designations? Presumably you used 16S sequencing, if so please provide primer sequences and the resulting 16S sequences. Perhaps deposit those 16S sequences in an online repository as appropriate.

- “Snodgrassella” is misspelled on line 84

- Figure 1B- the authors use broken y-axes to report spore counts. Broken axes often visually distort the magnitude of the difference between groups. Instead, these axes should be continuous (without breaks).

- Statistical test for Figure 1B is incorrect. For testing the means of many groups to each other, ANOVA or similar test is appropriate and should be performed first before pairwise tests of means. If performing many pairwise tests, the authors should discuss and report the method used to control for multiple testing.

- Figure 2H-O- What was the microbiota state of bees prior to administration of dsRNA nanoparticles? It is unclear if these are microbiota free bees or if they were colonized in some way. Either is fine, I think, but given the possible role of microbiota in Nosema disease identified in Figure 1. I would like to know.

- Figure 2N-O- Same comment as above, I recommend you do not use broken y-axes.

- Line 172. Plasmid pBTK519 confers kanamycin resistance, so I was confused to see Ampicillin mentioned for selection with this plasmid and used to pretreat bees before inoculation. Is this correct? Snodgrassella is very sensitive to Ampicillin, even in the presence of recombinant plasmids. Please confirm if Ampicillin was used when working with plasmid pBTK519.

- Figure 4 is overall very good, but I think it would benefit from an experimental schematic showing the timeline of experiment such as in 1A, 2H, and 3A. The assembly schematic in 4A is useful, but could do equally well as a supplemental figure. This is personal preference and offer merely as a recommendation that could make it easier for the average reader to interpret this figure.

- Figure 4H- same comment as above, I recommend you do not use broken y-axes. Especially since this scale is broken differently than in 1B and 2N-O. All of these panels (1B, 2N, 4H) report spore counts under relatively similar conditions, so a consistent, unbroken y-axis would make it easier for readers to interpret these results.

- Figure 4H- The authors report using a “least-significant difference” test, which is appropriate for comparing means of multiple groups after first performing an ANOVA to assess overall significance. Please report the use of ANOVA and appropriate test results, in addition to the LSD.

-Figure 5. The schematic shows dsRNA targeting only Nosema inside cells, but isn't it possible that the dsRNA acts on Nosema before it successfully infects cells?

-Line 278: genes are not “overrepresented” in the midgut. Should this read “ROS genes are overexpressed...”?

-Line 300: “bore” should be “core”

-Line 336: Snodgrassella does not “intensively” colonize the midgut. There is extensive evidence that *S. alvi* is primarily restricted to the ileum, though can be found in the midgut and rectum, especially of bees kept in the laboratory. The cited ref 35 even shows a very small amount of *S. alvi* in the midgut. This language should be softened to indicate that *S. alvi* can be found in the midgut.

-Line 771-772: Strain list repeats M0351

-Line 479: misspell “engineering”

-Lines 480-493: The authors should provide additional information regarding the design of dsRNA knockdown targets. Which honey bee and Nosema genomes were used to design the dsRNAs? Please include unique identifiers (Gene ID or Symbols) for all bee and nosema genes mentioned. These can be in Table S3/S4, or in the main text when the gene is first mentioned.

Reviewer #2 (Remarks to the Author):

Reviewers' Comments

Manuscript Number: NCOMMS-23-02016

Full Title: Engineered symbiotic bacteria interfering Nosema redox system inhibit microsporidia parasitism in honeybees

This is an interesting study, overall the information presented represents valuable information regarding the feasibility of using the engineered symbiotic bacteria interfering *Nosema redox* system inhibit microsporidia parasitism in honeybees. This paper has a potential to be accepted, but some important points have to be clarified or fixed

Here are the summarize of points which should be clarified or fixed:

Overall

1. Please check the units that the authors use in the manuscript (h or hours, min or minutes, μl or μL , ml or mL). Please use only one format following the instruction of the journal, then make it consistency throughout the manuscript.
2. All instruments should provide model, brand, country of origin.
3. Please explain the technique that the author used to selected engineered *S. alvi*
4. Do engineered *S. alvi* different from *S. alvi*? if yes, please explain the colony morphology, the grow rate or mechanism.
5. Why *S.alvi* has a high potential to reduce *N. ceranae* loads when compare with other microbe.
6. How do you apply to use engineered *S. alvi* to against *N. ceranae* in the colony level?

Introduction

7. Line 25, page 3: Please replace "Honey bees" with "Honeybees" and make it consistency throughout the manuscript.
8. Line 31, page 3: please replace "Varroa destructors" with "Varroa destructor".

Results and Discussions

The authors provide a very good results and discussions

Materials and Methods

9. Line 361, page 22: please provide more detail of incubator such as model, brand, country of origin.
10. Lines 363-367, page 22: please provide more detail how to grow the isolated gut bacteria. What is the condition (temperature, aerobic or anaerobic conditions, incubation period).
11. Line 364, page 22: please provide the full name of HIA and more detail of HIA (brand, country of origin).
12. Line 366, page 22: please provide the full name of MRS and more detail of MRS (brand, country of origin).

13. Line 370, page 22: please provide the size of axenic cup cages
14. Line 371, page 23: Please replace "24h" with "24 h"
15. Line 372, page 23: please provide the size of cup cage
16. Lines 371-372, page 23: Please use only one format of unit (24 h or 24 hours) and make it consistency throughout the manuscript.
17. Line 373, page 23: Please replace "colony-forming units" with "colony-forming units (CFU)"
18. Lines 374-377, page 23: What is the concentration of sucrose solution? Please provide the percentage of sucrose solution and make it consistency throughout the manuscript.
19. Lines 374-375, page 23: Do 1×PBS and sucrose solution sterilize? If yes, please replace "1×PBS" with "sterilized 1×PBS" and replace "sucrose solution" with "sterilized sucrose solution"
20. Line 376, page 23: Please replace "1 mL 1×PBS" with "1 mL of sterilized 1×PBS" and replace "sucrose solution" with "sterilized sucrose solution"
21. Line 377, page 23: Please replace "1mL sterilized sucrose solution" with "1mL of sterilized sucrose solution"
22. Lines 376-377, page 23: How the authors make sure the final concentration of bee gut bacterial strains is ~10⁸ CFU/mL, please explain the way that the author use.
23. Line 383, page 23: Please provide the number of Whatman filtering paper.
24. Line 390, page 23: Please provide more detail of hemocytometer such as brand, country of origin.
25. Line 401, page 24: Please provide more detail of microscope such as model, brand, country of origin.
26. Line 479, page 28: please replace "egineering" with "engineering"
27. Lines 497-498, page 29: please provide the name of media
28. Line 505, page 29: please replace "3824" with "3,824"

References

29. Lines 538, 545, 549, 555, page 31: please italic the genus and species name.
30. Lines 580, 590, 600, page 32: please italic the genus and species name.
31. Lines 629, 637-638, 640-641, 654-655, page 33: please italic the genus and species name.
32. Lines 668, 674, 681, 687, page 34: please italic the genus and species name.
33. Lines 702, 706, 709, 717, 724, 737, page 35: please italic the genus and species name.
34. Lines 754, 757, page 36: please italic the genus and species name.

Supplementary Information

35. Line 771, page 38: Please provide the full name of E. coli MFDpir

Point-by-point response

We appreciated all the comments from the reviewers. Compared with the previous version of manuscript, we have made mainly these modifications:

1. We have verified the colonization level of all isolated strains;
2. We have obtained a larger sample size to confirm the inhibition against *Nosema*;
3. We have tested the Duox and Nox gene regulation by other bee gut bacteria;
4. We have also verified the inhibition against *Nosema* by engineered strains using qPCR;
5. We have added more methodological information and discussion regarding the use of engineered strains in the field.

We have provided a point-by-point response to all suggestions.

All line numbers refer to those in the WORD file with “Review”→“Simple Markup” option.

REVIEWER COMMENTS

Reviewer #1 (Remarks to the Author):

In this work, Lang et al provide an important contribution to our understanding of how infection by the bee pathogen *Nosema ceranae* is affected by the bee microbiota and how engineered gut symbionts may be used to protect honey bees from this pathogen. The authors' findings support an important role for reactive oxygen species in honey bee resistance to *Nosema*, and suggest in turn that *Nosema* antioxidant systems are important for successful infection. Through a series of experiments with bees given a controlled microbiota the authors further link production of ROS in bee guts to the presence of *Snodgrassella alvi*, a core bee gut microbiome member. Finally, the authors build on recently published work and adapt a *Snodgrassella* dsRNA expression system, previously used to combat other bee pathogens, to directly target *Nosema* and demonstrate that colonization with these engineered bacteria can lower *Nosema* spore counts. Overall, this work provides useful insight into mechanisms of *Nosema* infection and demonstrates the repeatability and extensibility of the approach of using engineered *S. alvi* to produce dsRNA to protect bee health.

I commend the authors on this interesting and creative multidisciplinary work, and I believe it will be a useful contribution to the field. I do, however, have concerns with how some experiments were carried out and interpreted, and I believe there are several missing pieces of methodological information that are essential to fill in.

>>>Reply: We thank the reviewer who appreciate the significance of our work. We have performed additional experiments required by the reviewer and have added more the methodological information accordingly.

Major concerns:

-The findings reported in Fig 1. and discussed in lines 88-91 are over-interpreted compared to the evidence presented. First, the authors provide no evidence that the inoculation with their bacteria were successful. While bee gut bacteria generally colonize microbiota-free (MF) bees well, there is extensive strain to strain variation in colonization. These are newly isolated strains with unknown colonization dynamics. In order to claim a difference between *S. alvi* and the other bacteria, the authors should verify that these strains of bacteria can recolonize bees, or the observed differences could be due to lack of colonization. The authors can do this by reporting CFU counts from the guts of a sample of colonized bees compared to MF bees prior to inoculation.

>>>Reply: We thank the reviewer's constructive suggestions. We apologize that we might not make it clear enough. All the strains used in this work are acquired from our previous published paper and the colonization ability of some strains were reported: *Lactobacillus apis* W8172 (PMID 35440638); *Bifidobacterium chaladohabitans* W8113 (PMID 34784973, 35266810); *Bombilactobacillus mellis* W8089 (PMID 35266810). Nevertheless, we totally agree with the reviewer that it is necessary to verify the colonization level of these isolates. We did check the colonization of each strain before all subsequent experiments, which was not included in the previous manuscript. We have provided the CFUs of mono-colonized bees in **Supplementary Figure 1** now. It shows that all strains could re-colonize the MF bee guts successfully

(~10⁷ CFUs/gut).

Second, 3 bees per group is an unacceptably low sample size to draw robust conclusions for Fig 1B. As the authors do not report cupcage configuration for this experiment, I assume that each point is a single bee from the same cup. I encourage the authors to repeat the experiment shown in Fig 1B with a larger sample size (>= 5 bees from >= 2 independent cup cages per condition, or similar) to verify the findings.

>>>Reply: We thank the reviewer's suggestions. We have repeated this experiment with a larger sample size as suggested by the reviewer. Now we have 10 individual bees from two independent cup cages for each experiment condition (Fig. 1B, Supplementary Fig. 2).

- In lines 96-103 the authors appear to be making the connection between upregulation of Duox and Nox uniquely by *S. alvi* as contributing to its ability to lower *N. ceranae* infection. This interpretation would be strengthened if inoculation by other bacteria did not also trigger Duox and Nox during colonization. However, no data is shown on the possible upregulation of Duox and Nox by the other gut bacteria initially tested. I believe the authors could perform qPCR on bees colonized with the other bacteria for Duox and Nox, or explicitly mention that this upregulation of Duox and Nox may not be unique to *S. alvi*.

>>>Reply: We thank the reviewer for the constructive suggestion. We have performed qPCR to test the regulation of Duox and Nox genes by other gut members. It shows that other bee gut bacteria do not upregulate the gene expression of Duox or Nox, indicating that this upregulation is specific to *S. alvi*. We have added these results in Supplementary Fig. 3 and in the text (Line 102-103).

-The findings shown in Figure 2A-G and discussed in lines 124-137 require more methodological explanation in the main text and in the methods. It is unclear if the authors performed a new analysis that involved remapping and quantifying the previously published RNA-seq reads, or if the authors relied on the previous quantitation and plotted charts for their genes. If the former, the methods section should be updated to include all the computation steps involved (mapping, counting, etc). If the latter, the methods should still be clarified and relevant caveats with the original data, such as no biological replication, be specified.

>>>Reply: Thanks for the suggestion. We relied on the previous quantitation, since the raw sequencing data was not deposited publicly (the link in the paper has expired with no data on the webpage:

<https://www.ncbi.nlm.nih.gov/bioproject/PRJNA282511>). We have clarified in the figure legend (Line 785). The reviewer is correct that the original data was achieved without replication but the authors probed the gene expression profiles of 15 pooled bee individuals from three cup cages during the 6-day infection. We have clarified this point in the text (Line 116-118).

-Please provide more methodological details related to the microscopy shown in Figure 3D and discussed in 179-181. Include how the gut sections were prepared and whether they were fixed or imaged without fixation. Include the microscope, objective, and software used for collection of the images, as well as the wavelengths used for each channel and the exposure time for each image. Discuss if any adjustments were made to brightness for display. I am especially surprised to see the robust colonization of fluorescent *S. alvi* in the midgut, as there is usually substantial autofluorescence in the midgut that makes such visualization difficult. The images in Fig3D are also quite small and difficult to inspect closely- perhaps the authors could include full size images in the supplement or deposit raw images with an appropriate data repository.

>>>Reply: We thank the reviewer for this comment. We did not feed the bees pollen grains 24h before the dissection. We imaged without fixation. We used Leica SP8 Laser Confocal Microscope with a 20x objective. The LAS X software was used to collect photos using default parameters. For DAPI, we used 360 nm (excitation) and 460 nm (emission). For GFP, we used 488 and 511 nm, respectively. The exposure time is not an adjustable parameter for LAS X, Smart Gain is the control of image brightness. We used 700-720 for both channels as suggested by the user manual (https://health.usf.edu/-/media/Files/Medicine/Core/leica_sp8_users_manual.ashx?la=en&hash=E294E0A9AA76453DB9F3111022BA1A84907A799).

We have added more details in the method now (Line 493-500). Since one of the raw photos for the Fig. 3D was missed (*S. alvi* WT), we replaced another view from the same batch of experiment. We have also provided the original TIFF images as supplementary files for review.

Figure 4H, Lines 223-226, 253-254, and 344-350. I commend the authors for using a mixture of strains to deliver multiple dsRNAs, and it has a strong effect. However, repeatedly the authors claim that "inhibitory effect by a mixture of bacteria delivering all six dsRNA was better than the colonization with single strains" or similar. But the data in Fig 4H shows no significant difference between the mix and pDS-GS individually. Thus, the magnitude of knockdown seen with the mix could be primarily driven pDS-GS instead of some synergistic activity between them. This should be made clear in the text, and relevant lines changed indicate the mix did better than many, but not all, individual strains.

>>>Reply: Thanks for pointing this out. We totally agree with the reviewer. Now we have also quantified the *Nosema* load using qPCR. We have rephrased the relevant sentences to clarify this point (Line 192-200; 214-217; 297-300).

I have some concern that the only assay used to assess *Nosema* infection throughout is spore counts. Was there any effect on bee survival in these experiments? It could also be appropriate to utilize qPCR to assess *Nosema* burden for some experiments, as described by the COLOSS Bee Book methods on *Nosema*. Including an additional unbiased

estimator of *Nosema* burden would strengthen the claims made regarding the experiments in Figure 4.

>>>Reply: Thanks for the suggestion. We have now also used qPCR to assess *Nosema* burden as described in the COLOSS bee book. This has been added in Fig. 4 now and in the Methods (Line 501-511).

Minor concerns:

- In the introduction the authors should clearly introduce and reference previous efforts to use dsRNA to target *Nosema*, as there are many previous reports of this approach (See work by J. Evans and others). This need not take away the novelty of what they do, but is important context and helps justify why using dsRNA with *S. alvi* is reasonable

>>>Reply: We thank the reviewer for the suggestion. We have cited more literatures and introduced previous reports using dsRNA targeting *Nosema* (Line 67-71).

- Please provide more methodological details on the provenance of the bacterial strains in this work. How did you verify the species designations? Presumably you used 16S sequencing, if so please provide primer sequences and the resulting 16S sequences. Perhaps deposit those 16S sequences in an online repository as appropriate.

>>>Reply: We thank the reviewers for their suggestions. We apologize that our statement in **Supplementary Table 1** was not appropriate. All strains used in this study are from our previous publications and all their genomes have been obtained (Wu et al. 2021, doi:10.1186/s40168-021-01174-y). We have modified **Supplementary Table 1** and provided the accession numbers of the published genomes.

- "Snodgrassella" is misspelled on line 84

>>>Reply: Sorry for the typo. This has been corrected (Line 88).

- Figure 1B- the authors use broken y-axes to report spore counts. Broken axes often visually distort the magnitude of the difference between groups. Instead, these axes should be continuous (without breaks).

>>>Reply: We thank the reviewer for the suggestion. We have modified all figures with broken y-axes.

- Statistical test for Figure 1B is incorrect. For testing the means of many groups to each other, ANOVA or similar test is appropriate and should be performed first before pairwise tests of means. If performing many pairwise tests, the authors should discuss and report the method used to control for multiple testing.

>>>Reply: We thank the reviewer for pointing this out. In this figure now, we only tested the difference between the MF group and each mono-colonization group, but not pairwise comparison between all groups. We have clarified in the text (Line 94) and the figure legend (Line 773-774).

- Figure 2H-O- What was the microbiota state of bees prior to administration of dsRNA nanoparticles? It is unclear if these are microbiota free bees or if they were colonized in some way. Either is fine, I think, but given the possible role of microbiota in *Nosema* disease identified in Figure 1. I would like to know.

>>>Reply: We thank the reviewer for pointing this out. We used microbiota-free bees here. We have modified Fig. 2 now.

- Figure 2N-O- Same comment as above, I recommend you do not use broken y-axes.

>>>Reply: We have modified Fig. 2.

- Line 172. Plasmid pBTK519 confers kanamycin resistance, so I was confused to see Ampicillin mentioned for selection with this plasmid and used to pretreat bees before inoculation. Is this correct? *Snodgrassella* is very sensitive to Ampicillin, even in the presence of recombinant plasmids. Please confirm if Ampicillin was used when working with plasmid pBTK519.

>>>Reply: We sincerely thank the reviewer pointing this out. We apologize that we made a mistake with the plasmid number here. We did use ampicillin here but we used pBTK501 (Amp^R) with the ampicillin resistance marker but not the pBTK519. We have modified throughout the manuscript. We also noticed that *Snodgrassella* may be sensitive to ampicillin even with the BTK plasmids. As shown by Leonard et al. 2019 (PMID 29608282), pBTK501 could not be conjugated into strain PEB0171 but strain wkB2 could be transformed with pBTK501. We supposed that may be a result of strain variation. Thus, we did check the sensitivity of our strain M0351. As shown in figures below, the engineered strain M0351 grew on HIA with ampicillin (100 µg/ml) but the strain without the plasmid could not.

S. alvi M0351+pDS-GS

Wild type *S. alvi* M0351

- Figure 4 is overall very good, but I think it would benefit from an experimental schematic showing the timeline of experiment such as in 1A, 2H, and 3A. The assembly schematic in 4A is useful, but could do equally well as a supplemental figure. This is personal preference and offer merely as a recommendation that could make it easier for the average reader to interpret this figure.

>>>Reply: We thank the reviewer for the suggestion. We have moved the assembly schematic figure to Supplementary Fig. 4 now and have added a new experimental schematic in Fig. 4A.

- Figure 4H- same comment as above, I recommend you do not use broken y-axes. Especially since this scale is broken differently than in 1B and 2N-O. All of these panels (1B, 2N, 4H) report spore counts under relatively similar conditions, so a consistent, unbroken y-axis would make it easier for readers to interpret these results.

>>>Reply: Thanks for the suggestion. We have modified the Fig. 4 now.

- Figure 4H- The authors report using a "least- significant difference" test, which is appropriate for comparing means of multiple groups after first performing an ANOVA to assess overall significance. Please report the use of ANOVA and appropriate test results, in addition to the LSD.

>>>Reply: We thank the reviewer for the suggestion. We first assessed the variance among treatments thorough ANOVA ($p < 0.01$ for both spore counts and qPCR approaches). Then Brown-Forsythe and Welch test was used for multiple comparisons. We have added more information in the figure legend now (Line 810-812).

-Figure 5. The schematic shows dsRNA targeting only *Nosema* inside cells, but isn't it possible that the dsRNA acts on *Nosema* before it successfully infects cells?

>>>Reply: We thank the reviewer for the suggestion. We agree with the reviewer and have modified the figure.

-Line 278: genes are not "overrepresented" in the midgut. Should this read "ROS genes are overexpressed..."?

>>>Reply: We have rephrased the sentence on Line 232.

-Line 300: "bore" should be "core"

>>>Reply: Corrected (Line 253).

-Line 336: Snodgrassella does not "intensively" colonize the midgut. There is extensive evidence that *S. alvi* is primarily restricted to the ileum, though can be found in the midgut and rectum, especially of bees kept in the laboratory. The cited ref 35 even shows a very small amount of *S. alvi* in the midgut. This language should be softened to indicate that *S. alvi* can be found in the midgut.

>>>Reply: We thank the reviewer pointing this out. We have rephrased the sentence (Line 288-289).

-Line 771-772: Strain list repeats M0351

>>>Reply: Modified (Supplementary Table 1).

-Line 479: misspell "engineering"

>>>Reply: Corrected (Line 444).

-Lines 480-493: The authors should provide additional information regarding the design of dsRNA knockdown targets. Which honey bee and *Nosema* genomes were used to design the dsRNAs? Please include unique identifiers (Gene ID or Symbols) for all bee and *nosema* genes mentioned. These can be in Table S3/S4, or in the main text when the gene is first mentioned.

>>>Reply: Thanks for the suggestion. We have added this information in Supplementary Table 3 now.

Reviewer #2 (Remarks to the Author):

This is an interesting study, overall the information presented represents valuable information regarding the feasibility of using the engineered symbiotic bacteria interfering *Nosema* redox system inhibit microsporidia parasitism in honeybees. This paper has a potential to be accepted, but some important points have to be clarified or fixed

Here are the summarize of points which should be clarified or fixed:

1. Please check the units that the authors use in the manuscript (h or hours, min or minutes, μ l or μ L, ml or mL). Please use only one format following the instruction of the journal, then make it consistency throughout the manuscript.

>>>Reply: We have modified the format of the units throughout the manuscript.

2. All instruments should provide model, brand, country of origin.

>>>Reply: We have provided all these information in the Methods now.

3. Please explain the technique that the author used to selected engineered *S. alvi*

>>>Reply: To obtain engineered bee gut *S. alvi*, we followed the the Functional Genomics Using Engineered Symbionts (FUGUES) procedure by Lariviere *et al.* published in *Naure Protocol* (PMID 36460809). We used the auxotrophic *E. coli* MFD_{pir} as donors to facilitate efficient postmating donor counterselection. *E. coli* MFD_{pir} is a Mu-free donor strain constructed by removal of the Mu prophages and mutation of *oriT*_{chrRP4} from the K-12 lineage (doi:10.1128/JB.00621-10). Specifically, MFD_{pir} is a diaminopimelic acid auxotroph and *recA* deletion mutant. Thus, we can select the engineered *S. alvi* by plating the conjugation mixtures (MFD_{pir} and *S. alvi*) onto selective plates with appropriate antibiotics but without diaminopimelic acid. We have added more details in the Methods (Line 459-472).

4. Do engineered *S. alvi* different from *S. alvi*? if yes, please explain the colony morphology, the grow rate or mechanism.

>>>Reply: Thanks for pointing this out. We did not observe any morphological or physiological difference between engineered strains and wild type strains of *Snodgrassella*. This was also reported by the Moran Lab in their previous papers (e.g., PMID 36460809, 29608282). We used the same genetic toolkit as they did. We have clarified this point in the text (Line 151-152)

5. Why *S.alvi* has a high potential to reduce *N. ceranae* loads when compare with other microbe.

>>>Reply: Thanks for pointing this out. This is also mentioned by Reviewer #1. We hypothesized that only *Snodgrassella* increased the gene expression of DUOX and

NOX specifically. We have added additional experiments required by Reviewer #1 to show that other gut bacteria did not affect these genes' expression in the honeybees (**Supplementary Fig. 3**).

6. How do you apply to use engineered *S. alvi* to against *N. ceranae* in the colony level?

>>>Reply: We thank the reviewer pointing this out. We did not used the engineered strains in the bee colony due to the ethical issue of gene escape. This needs to be addressed before engineered bacteria are applied to honeybees in the field. But we do agree with the reviewer that we may consider alternative strategies for the application of engineered strains. We are working on an auxotrophic gut strain which may promise its restriction to the gut environment, and this will be included in a separate work. We have added more discussions on Line 304-311.

7. Line 25, page 3: Please replace "Honey bees" with "Honeybees" and make it consistency throughout the manuscript.

>>>Reply: We have modified the words throughout the manuscript.

8. Line 31, page 3: please replace "Varroa destructors" with "Varroa destructor".

>>>Reply: Replaced (Line 31)

Results and Discussions

The authors provide a very good results and discussions

Materials and Methods

9. Line 361, page 22: please provide more detail of incubator such as model, brand, country of origin.

>>>Reply: We have added this information in the text (Line 317).

10. Lines 363-367, page 22: please provide more detail how to grow the isolated gut bacteria. What is the condition (temperature, aerobic or anaerobic conditions, incubation period).

>>>Reply: Thanks for the suggestions. We have added more details in the Methods (Line 319-326).

11. Line 364, page 22: please provide the full name of HIA and more detail of HIA (brand, country of origin).

>>>Reply: We have added this information (Line 321).

12. Line 366, page 22: please provide the full name of MRS and more detail of MRS (brand, country of origin).

>>>Reply: We have added this information (Line 324).

13. Line 370, page 22: please provide the size of axenic cup cages

>>>Reply: We have added this information (Line 328).

14. Line 371, page 23: Please replace "24h" with "24 h"

>>>Reply: Corrected (Line 330).

15. Line 372, page 23: please provide the size of cup cage

>>>Reply: Added (Line 331).

16. Lines 371-372, page 23: Please use only one format of unit (24 h or 24 hours) and make it consistency throughout the manuscript.

>>>Reply: We have edited the unit format throughout the manuscript (Line 331).

17. Line 373, page 23: Please replace "colony- forming units" with "colony-forming units (CFU)"

>>>Reply: Corrected (Line 332).

18. Lines 374-377, page 23: What is the concentration of sucrose solution? Please provide the percentage of sucrose solution and make it consistency throughout the manuscript.

>>>Reply: We used 50% (wt/vol) of sucrose solution. We have added this information in the text (Line 334-338).

19. Lines 374-375, page 23: Do 1×PBS and sucrose solution sterilize? If yes, please replace "1×PBS" with "sterilized 1×PBS" and replace "sucrose solution" with "sterilized sucrose solution"

>>>Reply: The reviewer is correct. We have modified the text (Line 333).

20. Line 376, page 23: Please replace "1 mL 1×PBS" with "1 mL of sterilized 1×PBS" and replace "sucrose solution" with "sterilized sucrose solution"

>>>Reply: We have modified the text (Line 336).

21. Line 377, page 23: Please replace "1mL sterilized sucrose solution" with "1mL of sterilized sucrose solution"

>>>Reply: Corrected (Line 336).

22. Lines 376-377, page 23: How the authors make sure the final concentration of bee gut bacterial strains is $\sim 10^8$ CFU/mL, please explain the way that the author use.

>>>Reply: We quantified the concentration of the bacteria suspension by plate count. We have added more details in the text (Line 337).

23. Line 383, page 23: Please provide the number of Whatman filtering paper.

>>>Reply: We have provided this information (Line 344).

24. Line 390, page 23: Please provide more detail of hemocytometer such as brand, country of origin.

>>>Reply: We have provided this information (Line 353).

25. Line 401, page 24: Please provide more detail of microscope such as model, brand, country of origin.

>>>Reply: We have provided this information (Line 364).

26. Line 479, page 28: please replace "egineering" with "engineering"

>>>Reply: Corrected (Line 444).

27. Lines 497-498, page 29: please provide the name of media

>>>Reply: We have provided this information (Line 461).

28. Line 505, page 29: please replace "3824" with "3,824"

References

>>>Reply: Corrected (Line 474).

29. Lines 538, 545, 549, 555, page 31: please italic the genus and species name.

>>>Reply: Corrected accordingly.

30. Lines 580, 590, 600, page 32: please italic the genus and species name.

>>>Reply: Corrected accordingly.

31. Lines 629, 637-638, 640-641, 654-655, page 33: please italic the genus and species name.

>>>Reply: Corrected accordingly.

32. Lines 668, 674, 681, 687, page 34: please italic the genus and species name.

>>>Reply: Corrected accordingly.

33. Lines 702, 706, 709, 717, 724, 737, page 35: please italic the genus and species name.

>>>Reply: Corrected accordingly.

34. Lines 754, 757, page 36: please italic the genus and species name.

>>>Reply: Corrected accordingly.

Supplementary Information

35. Line 771, page 38: Please provide the full name of E. coli MFDpir

>>>Reply: Modified (Supplementary Table 1).

REVIEWERS' COMMENTS

Reviewer #1 (Remarks to the Author):

The authors have effectively addressed all of my concerns. This work is an impressive contribution to the field.

There are two typos in the updated figure 5:

Misspelled "snodgrassella"

Misspelled "thioredoxin"

Reviewer #2 (Remarks to the Author):

Dear authors

All of responded comments is very clear. This study should be accepted for publication.

Point-by-point response

REVIEWER COMMENTS

Reviewer #1 (Remarks to the Author):

-The authors have effectively addressed all of my concerns. This work is an impressive contribution to the field.

Reply: We thank the reviewer for the constructive suggestions.

-There are two typos in the updated figure 5:

Misspelled "snodgrassella"

Misspelled "thioredoxin"

Reply: Corrected on the new Figure 5.

Reviewer #2 (Remarks to the Author):

-All of responded comments is very clear. This study should be accepted for publication.

Reply: We thank the reviewer for providing valuable comments.